# Remitting Seronegative Symmetrical Synovitis with Pitting Edema Syndrome Worsen after the Administration of Dulaglutide

**DOI:** 10.3390/medicina58020289

**Published:** 2022-02-14

**Authors:** Toshitaka Sawamura, Shigehiro Karashima, Ai Ohmori, Kei Sawada, Mitsuhiro Kometani, Yoshiyu Takeda, Takashi Yoneda

**Affiliations:** 1Asanogawa General Hospital, 83 Kosakamachi, Kanazawa 910-8621, Japan; cats2ai@yahoo.co.jp (A.O.); iekadawas0212@yahoo.co.jp (K.S.); takeday20721@yahoo.co.jp (Y.T.); 2Department of Endocrinology and Metabolism, Kanazawa University Graduate School of Medicine, 13-1 Takaramachi, Kanazawa 920-8641, Japan; kometankomekome@yahoo.co.jp (M.K.); endocrin@med.kanazawa-u.ac.jp (T.Y.); 3Department of Health Promotion and Medicine of the Future, Kanazawa University, 13-1 Takaramachi, Kanazawa 920-8641, Japan; hypertension14090@yahoo.co.jp

**Keywords:** RS3PE syndrome, GLP-1 analogue, dulaglutide

## Abstract

Remitting seronegative symmetrical synovitis with pitting edema (RS3PE) syndrome is characterized by symmetrical polyarthritis and limb pitting edema. Although the detailed mechanisms of this syndrome have not been clearly understood, some agents including dipeptidyl peptidase-4 inhibitors have been reported to induce RS3PE syndrome. However, glucagon-like peptide-1 (GLP-1) analogues have not been reported to be associated with this syndrome. A 91-year-old woman was admitted to our hospital with complaints of severe polyarthritis and limb edema. She was diagnosed with RS3PE syndrome. Oral prednisolone improved her symptoms. However, her symptoms worsened after the administration of dulaglutide, with elevated serum inflammatory markers. Discontinuation of dulaglutide without additional treatment improved her symptoms and laboratory findings. This case might indicate the possibility of development and worsening of RS3PE syndrome caused after GLP-1 analogue.

## 1. Introduction

Remitting seronegative symmetrical synovitis with pitting edema (RS3PE) syndrome is characterized by elderly-onset symmetrical polyarthritis with pitting edema of the limb. Olive et al. described the sudden onset of polyarthritis, bilateral pitting edema of both hands and feet, seronegative for rheumatoid factor (RF), and age >50 years as the diagnosis criteria of this syndrome [1]. Some are idiopathic, but others are associated with malignant tumors, dipeptidyl peptidase-4 (DPP-4) inhibitors, and insulins [2,3,4]. However, other anti-diabetic agents, including glucagon-like peptide-1 (GLP-1) analogues, have not been reported to be related to this condition. Herein, we report a case of RS3PE syndrome in which polyarthritis and pitting edema remarkably worsened after the administration of dulaglutide. Clinical findings of RS3PE syndrome were improved after the discontinuation of dulaglutide.

## 2. Case Report

A 91-year-old Japanese woman admitted to our hospital with complaint of severe joints pain and pitting edema of her limb. She underwent surgery for ascending colon cancer 11 years ago and has not had any recurrence. She had been treated for type 2 diabetes and hypertension with the following medications: telmisartan at a dose of 40 mg/day, voglibose at a dose of 0.6 mg/day, and insulin degludec at a dose of 10 units/day. Her insulin had been injected by her son and his wife. These medications had not been changed in the previous 12 months. She had no significant family history of diseases, including rheumatic disorders.

She had experienced severe joints pain in bilateral wrists and fingers 7 weeks before hospitalization. The joint pain spread to her ankle joints within a few days. These symptoms worsened to maximum intensity within a week, and she could barely walk. Five weeks before her hospitalization, her daughter noticed that she had pitting edema of her limb. Three weeks before hospitalization, she visited her family clinic and received diuretics and painkillers. However, neither her pain nor edema improved, and she was admitted to our hospital.

Her height, weight, and body mass index were, 146 cm, 46.2 kg, and 21.7 kg/m^2^, respectively. Her vital signs at hospitalization were as follows: blood pressure of 120/84 mmHg, heart rate of 78 beats/min, temperature of 37.4 °C, and oxygen saturation of 99% in room air. Examination showed swelling in her bilateral wrist, fingers, and ankle joints. As shown in Figure 1, she had severe slow pitting edema in her limb. As shown in Table 1, a laboratory evaluation showed elevated serum C-reactive protein (CRP), erythrocyte sedimentation rate (ESR), and matrix metalloproteinase-3 levels. Tests for RF and anti-cyclic citrullinated peptide antibodies were negative. Slightly elevated brain natriuretic peptide level was observed. However, ultrasound cardiography revealed no dilation of the inferior vena cava (IVC) and no loss of respiratory change in IVC. Musculoskeletal ultrasound showed thickening of the bicep tendon and increased blood flow to the area. However, bone erosion was not observed both in musculoskeletal ultrasound and X-rays. Systemic computed tomography and upper- and lower-gastrointestinal endoscopy showed no evidence of malignant tumor.

Her clinical course is shown in Figure 2. She was diagnosed with RS3PE syndrome and treated with oral prednisolone 15 mg/day. At the same time prednisolone was given, esomeprazole 20 mg/day and alendronate sodium hydrate 35 mg/week were administrated to prevent side effects of prednisolone. The treatment improved her joints pain and edema of the limb with normalization of CRP and ESR levels within a week. After 2 weeks of treatment with prednisolone 15 mg/day, the dose of prednisolone was reduced to 12.5 mg/day. A total of 3 weeks after prednisolone treatment was started, insulin degludec was switched to dulaglutide 0.75 mg/week. This change in treatment was made to reduce the injection times by her family and to ease the burden of her family. Three days after the first injection of dulaglutide, her joint pain and edema of the limb flared up. A laboratory examination showed re-elevated serum CRP and ESR levels. Thus, the patient was switched from dulaglutide to insulin degludec. Two weeks after the discontinuation of dulaglutide, her polyarthritis and pitting edema of the limb disappeared without additional treatments. Her polyarthritis and edema did not worsen, and the dose of prednisolone was reduced to 10 mg/day and she was discharged. After being discharged, she continued to visit our hospital once a month and was monitored for her symptoms and blood examinations. Her symptoms and inflammatory markers in blood examinations did not worsen, and the dose of oral prednisolone was gradually reduced. At 12 months after discharge, her oral prednisolone had been decreased to a dose of 2 mg/day. She had no evidence of bone erosion on musculoskeletal ultrasound and X-rays for 12 months.

## 3. Discussion

To our knowledge, this is the first case of the worsening of RS3PE syndrome following the administration of GLP-1 analogue, dulaglutide.

The first important discussion point is about the diagnosis, as the diagnosis of RS3PE syndrome is a diagnosis of exclusion. The differentiation from elderly-onset rheumatoid arthritis (EORA) is of utmost importance. Physical characteristics of EORA resemble RS3PE syndrome. EORA is often acute-onset, relatively lower positive rates of RF and anti-CCP antibody compared to younger onset rheumatoid arthritis, and responsive to low dose steroid. Lack of bone erosion on X-rays and musculoskeletal ultrasound does not deny the possibility of EORA because the abnormalities are not apparent at the initial stages. However, bone erosion had not been observed on X-rays and musculoskeletal ultrasound for 12 months in our case. This course is more suspicious of RS3PE syndrome. Furthermore, we cannot fully exclude the possibility of heart failure as a cause of edema in extremities in our case. This patient had old age, hypertension, and type 2 diabetes, and revealed an elevated BNP level. Therefore, the possibility remains that she was complicated by mild heart failure that could not be detected by echocardiography. Furthermore, heart failure may improve just after an introduction of steroid since steroids can improve the hypersecretion of antidiuretic hormone. However, we do not think that her symptoms could be explained by heart failure alone. This is because exacerbation of edema was accompanied by peripheral arthritis and elevated inflammatory markers. These symptoms are not observed in patients with heart failure. Based on this differentiation, we diagnosed the patient with RS3PE syndrome and its relapse after the administration of dulaglutide. The adverse reaction of dulaglutide was consider probable according to World health Organization-Uppsala Monitoring Centre scale, based on temporality, resolution on withdrawal of the drug, etc. [5].

GLP-1 analogues and DPP-4 inhibitors are similar in terms of elevating serum GLP-1 levels and insulin levels. DPP-4 inhibitors are anti-diabetic drugs that inhibit the DPP-4 enzyme that breaks down GLP-1 and gastric inhibitory polypeptide (GIP), which is reported to be 1 of the causes of RS3PE syndrome. An increase in GLP-1 and GIP results in the stimulation of insulin release and inhibition of glucagon release, thereby lowering blood glucose levels. Administration of DPP-4 inhibitors increases stromal cell-derived factor-1 (SDF-1) level by inhibiting DPP-8 and DPP-9, similar to DPP-4 [6]. Elevated SDF-1 level increases leukocyte migration and serum vascular endothelial growth factor (VEGF) level [7]. The increase in serum VEGF level induces and worsens RS3PE syndrome [8]. This is the mechanism in which DPP-4 inhibitors induce RS3PE syndrome. On the other hand, GLP-1 analogues do not increase SDF-1 levels because they do not act on DPP-4, DPP-8, and DPP-9. However, the administration of GLP-1 analogue might affect serum VEGF level in a manner independent of SDF-1. Xie et al. reported that GLP-1 enhances the proliferation and differentiation of endothelial progenitor cells via upregulation of VEGF in vivo [9]. Further, Sato K. et al. reported that liraglutide, one of GLP-1 analogues, increases VEGF level in brain tissue and decreases infarct volume in rats that received middle cerebral artery occlusion [10]. This GLP-1-induced increase in VEGF level might be a mechanism by which dulaglutide promoted the occurrence and worsening of RS3PE syndrome. However, the serum VEGF level was not evaluated in our case.

Mainali et al. reported a case of RS3PE syndrome caused by insulin therapy [4]. Furthermore, insulin has been reported to increase VEGF mRNA expression and VEGF levels in cardiomyocytes and kidney podocytes [11,12]. Thus, they referred to the possibility that insulin-induced elevation of VEGF may induce RS3PE syndrome. Moreover, Zhang Q. et al. reported that the elevation of blood glucose levels increases serum VEGF level [13]. Therefore, switching from degludec to dulaglutide can increase serum VEGF level and worsen RS3PE syndrome if the blood glucose level increases. In our case, fasting blood glucose level evaluated by self-monitoring of blood glucose level was not elevated after the administration of dulaglutide as shown in Figure 2. However, steroid-induced diabetes does not necessarily show elevated fasting blood glucose levels but is well-characterized by elevated postprandial blood glucose levels. Unfortunately, more detailed information on clinical course of glycemic control including postprandial blood glucose levels on this patient were not available. Therefore, it cannot be excluded that RS3PE syndrome relapsed in condition with worsening of blood glucose control due to the administration of steroid and the switch from degludec to dulaglutide.

## 4. Conclusions

We report the first case of RS3PE syndrome that worsened after the administration of GLP-1 analogue, dulaglutide This case is significant for understanding the possibility of GLP-1 analog-induced development and exacerbation of RS3PE syndrome; further studies are needed to clarify the mechanism of GLP-1 analog-induced RS3PE syndrome.

## Figures and Tables

**Figure 1 medicina-58-00289-f001:**
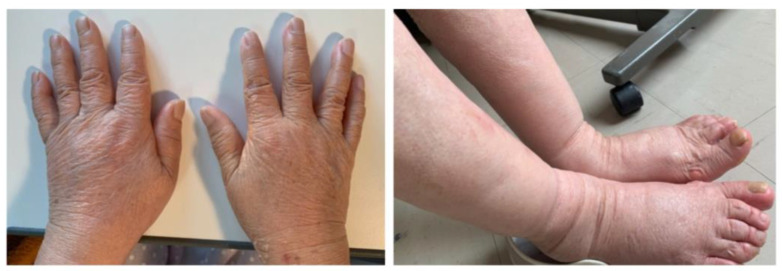
Photograph of the patient’s limb. Severe pitting edema of the limb was observed.

**Figure 2 medicina-58-00289-f002:**
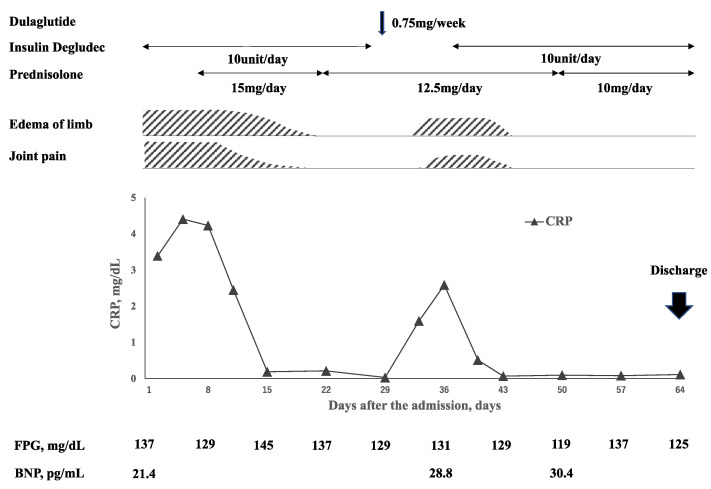
Time course of clinical symptoms, inflammatory markers, fasting plasma glucose (FPG), and brain natriuretic peptide (BNP) levels. Polyarthritis and edema were improved after the administration of oral prednisolone at a dose of 15 mg/day, and the prednisolone dose was decreased to 12.5 mg/day. At 7 days after the dose reduction, insulin degludec was switched to dulaglutide. Fasting blood glucose level described as an average over a week was not increased after the switching to dulaglutide. At 3 days after dulaglutide administration, her polyarthritis and edema worsened with elevated CRP level. Discontinuation of dulaglutide improved these symptoms and inflammatory markers.

**Table 1 medicina-58-00289-t001:** Laboratory examination on admission.

Variable	Value	Normal Range
<Blood>		
WBC count,/μL	8100	4800–10,800
RBC count, ×10^6^/μL	2.68	4.20–5.40
Hemoglobin, g/dL	8.6	12.0–16.0
Platelet count, ×10^3^/μL	278	130–400
TP, g/dL	6.5	6.7–8.3
Albumin, g/dL	3.3	4.0–5.0
γGTP, U/L	12	10–47
Total protein, g/dL	6.5	6.7–8.3
Albumin, g/dL	3.3	4.0–5.0
AST, U/L	24	13–33
ALT, U/L	20	6–22
γGTP, U/L	12	10–47
BUN, mg/dL	13	8.0–22.0
Creatinine, mg/dL	0.71	0.50–0.80
eGFR, mL/min/	57.0	60–90
CRP, mg/dL	3.39	<0.3
FPG, mg/dL	114	69–109
HbA1c, %	6.7	4.6–6.2
TSH, μIU/mL	1.23	0.34–3.88
FT4, ng/mL	1.12	0.95–1.74
BNP, pg/mL	21.4	<18.4
D-Dimer, μg/mL	0.5	<1.0
ESR, mm/h	110	<10
MMP-3, ng/mL	468	17.3–59.7
RF, IU/mL	<3.0	<15
Anti-CCP antibody, U/mL	<4.5	<4.5
<Urine>		
Protein	negative	negative
Occult blood	negative	negative

WBC, white blood cells; RBC, red blood cells; TP, total protein; AST, aspartate aminotransferase; ALT, alanine aminotransferase; γGTP, γ-glutamyltranspeptidase; BUN, blood urea nitrogen; eGFR; estimated glemerular filtration rate; CRP, C-reactive protein; Ig, immunoglobin; FPG, fasting plasma glucose; HbA1c, hemoglobin A1c; TSH, thyroid stimulating hormone; FT4, free thyroxine; BNP, brain natriuretic peptide; ESR, MMP-3, matrix metalloproteinase-3; erythrocyte sedimentation rate; RF, rheumatoid factor; anti-CCP-Ab, anti-citrullinated peptides antibody.

## Data Availability

The original contributions presented in the study are included in the article. Further inquiries can be directed to the corresponding author.

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
