# Peer review of "Remitting Seronegative Symmetrical Synovitis with Pitting Edema Syndrome Worsen after the Administration of Dulaglutide"

_medicina, 2022, doi:10.3390/medicina58020289_

Round 1

Reviewer 1 Report

The authors described a case of RS3PE which relapsed at switching insulin to GLP-1 analogue. The case report is of interest, however, I would like the authors to show more solid evidence to reach their conclusion considering RS3PE is a diagnosis of exclusion.

Major concerns

  1. A possibility of heart failure as a cause of edema in extremities: The patient was in a very old age, had hypertension and type 2 DM, and revealed an elevated BNP level (Table 1). Therefore, it is likely the patient had heart failure. Furthermore, heart failure may improve just after an introduction of steroid because steroid may improve the hypersecretion of ADH. Chest X rays and ultrasound cardiography findings before the onset and/or at the relapse of RS3PE, and transition of BNP levels after the introduction of steroid should be shown.
  2. Differential diagnosis of elderly-onset rheumatoid arthritis (EORA): Physical characteristics of EORA resembles RS3PE. EORA is often acute-onset, relatively lower positive rates of RF and anti-CCP Ab compared to younger onset RA, and responsive to low dose steroid. Lack of bone erosion and joint space narrowing on X-rays does not deny a possibility of RA because the abnormalities are not apparent at an initial stage. It is desirable to show tenosynovitis images compatible with RS3PE on MRI and/or musculoskeletal ultrasounds.
  3. Glycemic control and RS3PE: The reason why the authors decided switching insulin to GLP-1 analogue is not clear. Other than anti-diabetic drugs, DM itself was reported to be related to onset of RS3PE. Therefore, it is plausible RS3PE relapsed in conjunction with worsening of glycemic control due to the switch of anti-diabetic drugs. Steroid-induced DM does not necessarily show elevated fasting blood glucose levels, but is well-charecterized by elevated postprandial blood glucose levels. Therefore, the authors should show more detailed information on clinical course of glycemic control including postprandial blood glucose levels in Figure 2.

Minor concerns

  1. Was the pitting edema slow or fast edema?
  2. An elevated MMP-3 level is associated with kidney dysfunction. Considering the patient’s age, existence of CKD was highly likely even her serum creatinine level was within normal ranges. It is desirable to provide the patient’s body weight and calculate estimated GFR.

Author Response

Thank you for your thoughtful and constructive suggestions you provide regarding our manuscript. We are delighted to hear that you think our work will spark debate in our field. In the attached file, you will find our responses to each of your points and suggestions. In addition to your suggestion, we have corrected two errors. The first is miss typing error about dulaglutide (Line110). the second is the entering wrong test results in Table1. We modified the level of RF 8.1 to <3.0.

We are grateful for the time and energy you expended on our work.

Reviewer 2 Report

  1. Describe the meaning of RF the first time the authors neme this term  in point 2.
  2. Can the authors mention the interval time between other drugs that were introduced with the start of the RS3PE?
  3. Table 1: show heigh, size and BMI

Author Response

Thank you for your thoughtful and constructive suggestions you provide regarding our manuscript. We are delighted to hear that you think our work will spark debate in our field. In the attached file, you will find our responses to each of your points and suggestions. In addition to your suggestion, we have corrected two errors. The first is miss typing error about dulaglutide (Line103). the second is the entering wrong test results in Table1. We modified the level of RF 8.1 to <3.0

Round 2

Reviewer 1 Report

(There are no comments)

Author Response

 Thank you for your thoughtful suggestions for our manuscript. We are grateful for the time and energy you expended on our work. In your review report (round2), no comment were described. I am attaching the revised version that I submitted last time once. I would appreciate it if you could let me know if there is anything that needs to be corrected to make our article better.
